# Changes in Phytohormones and Transcriptomic Reprogramming in Strawberry Leaves under Different Light Qualities

**DOI:** 10.3390/ijms25052765

**Published:** 2024-02-27

**Authors:** Peng Li, Zhiqiang Wang, Xiaodi Wang, Fengzhi Liu, Haibo Wang

**Affiliations:** Institute of Pomology of CAAS, Xingcheng 125100, China; lipeng05@caas.cn (P.L.); wangzhiqiang01@caas.cn (Z.W.); wangxiaodi@caas.cn (X.W.)

**Keywords:** strawberry leaves, light quality, phytohormones, transcriptomic reprogramming

## Abstract

Strawberry plants require light for growth, but the frequent occurrence of low-light weather in winter can lead to a decrease in the photosynthetic rate (Pn) of strawberry plants. Light-emitting diode (LED) systems could be used to increase Pn. However, the changes in the phytohormones and transcriptomic reprogramming in strawberry leaves under different light qualities are still unclear. In this study, we treated strawberry plants with sunlight, sunlight covered with a 50% sunshade net, no light, blue light (460 nm), red light (660 nm), and a 50% red/50% blue LED light combination for 3 days and 7 days. Our results revealed that the light quality has an effect on the contents of Chl a and Chl b, the minimal fluorescence (F_0_), and the Pn of strawberry plants. The light quality also affected the contents of abscisic acid (ABA), auxin (IAA), *trans*-zeatin-riboside (*t*Z), jasmonic acid (JA), and salicylic acid (SA). RNA sequencing (RNA-seq) revealed that differentially expressed genes (DEGs) are significantly enriched in photosynthesis antenna proteins, photosynthesis, carbon fixation in photosynthetic organisms, porphyrin and chlorophyll metabolisms, carotenoid biosynthesis, tryptophan metabolism, phenylalanine metabolism, zeatin biosynthesis, and linolenic acid metabolism. We then selected the key DEGs based on the results of a weighted gene co-expression network analysis (WGCNA) and drew nine metabolic heatmaps and protein–protein interaction networks to map light regulation.

## 1. Introduction

Strawberry (*Fragaria* × *ananassa* Duch.) plants are forced to endure a variety of adverse environmental conditions; among these, the low-light condition is one of the main factors affecting the fruit yield and quality of strawberry plants. Previous studies have shown that the intensity of the light induces changes in the phenotypic characteristics of strawberry plants through respiration, photosynthesis, and low-light conditions [1]. Light-emitting diode (LED) systems enhance the efficiency of photosynthesis in strawberry plants and their fruit quality and yield [2]. Blue/red (1:1) light and white/yellow (400–700 nm) light have greater effects on the reproductive traits of strawberry plants than blue light (460 nm) and red light (660 nm) [3]. The maximum performance and biomass production of strawberry plants are obtained when using a prolonged and high-intensity light with a red-light component [4].

The photosynthetic rate (Pn) of strawberry leaves under red light treatment (660 nm, 80 μmol·m^−2^·s^−1^) is higher than that under blue light treatment (450 nm, 80 μmol·m^−2^·s^−1^), but blue light promotes flowering to a greater extent than red light [5]. Blue light regulates early flowering through the *FaBBX29* transcription factor (a BBX family gene) [6] and enhances the fruit set by approximately 25% compared to red light in protected strawberry cultivation systems [7]. Furthermore, the *FvFT1* (flowering locus *t*) gene advances flowering under far-red and blue light treatments to a greater extent than red light and short-day treatments, and the homolog of terminal flower 1 (*FvTFL1*) is specifically required by the plants to respond to blue light [8].

Moreover, blue light is able to improve the anthocyanin content of fruit to a greater extent than red light [9], and a 50% blue/50% red LED light combination significantly promotes the biosynthesis of anthocyanin and proanthocyanidin in two strawberry genotypes during fruit development [10]. Red light contributes to the synthesis of proanthocyanidins [11], and the blue light signal transduction module FaCRY1-FaCOP1-FaHY5 regulates the accumulation of anthocyanin. A co-expression network analysis of differentially expressed metabolites (DEMs) and differentially expressed genes (DEGs) also revealed that blue light can co-upregulate the synthesis of chlorogenic acid and *hydroxycinnamoyl-CoA:shikimate hydroxycinnamoyltransferase* (*FvHCT*) gene expression compared with red light [12].

Phytohormones participate in light response processes. For example, blue and red light treatments significantly increase the content of indole-3-acetic acid (IAA) in fruit, and a treatment comprising red/blue (100 μmol·m^−2^·s^−1^) + 100 mM of sucrose promotes the detached ripening of strawberries by regulating the signalings of abscisic acid (ABA) and IAA [13]. Light and ABA independently regulate *FaMYB10* (an *R2R3 MYB* transcription factor) [14] and *FaWRKY71* (*WRKY* DNA-binding protein 71) [15] during the ripening of strawberry fruits. Furthermore, a study has suggested that a treatment comprising blue light (BL) + salicylic acid (SA) maintains the postharvest quality of strawberry fruits during refrigeration [16]. In addition, SA enhances the accumulation of proanthocyanidins and the upregulation of pathogenesis-related genes and induces resistance against *Podosphaera aphanis* in strawberry plants by antagonizing with jasmonic acid (JA) [17]. Treatment with methyljasmonate (MeJA) appears to alter the metabolism of strawberry plants, thus enhancing the ability of the tissue to withstand water stress [18]. In summary, LED systems are frequently used to promote the early flowering of strawberry plants and enhance their production and quality; however, the key genes involved in Pn are still unclear.

In this study, we treated strawberry plants with sunlight (control group, CK); sunlight covered with a 50% sunshade net (low-intensity light, LL); no light (NL); blue light (BL, 460 nm); red light (RL, 660 nm); and a 50% blue/50% red LED light (RBL) combination for 3 days and 7 days. First, we collected strawberry leaves and detected the changes in the relative water content (RWC), photosynthetic pigment content, Pn, and minimal fluorescence (F_0_). Then, we analyzed the phytohormone species that were present, screened the key genes regulated by different light qualities based on the results of the transcriptomic reprogramming, and created a protein–protein interaction network map of the light regulation observed in the strawberry leaves. Finally, we found that there are 10 proteins associated with hormone synthesis and signal transduction, 11 proteins associated with photosynthesis, 6 proteins associated with chlorophyll metabolism, 9 proteins associated with photosynthesis antenna proteins, and 10 proteins associated with carbon fixation in photosynthetic organisms participating in light regulation.

## 2. Results

### 2.1. Analysis of Physiological Indicators

The RWC is an important physiological indicator in photosynthesis and carbon fixation. In this study, the RWCs of the low-intensity light group for 3 days (LL1) and the no-light group for 3 days (NL1) were increased compared with that of the control group for 3 days (CK1) in strawberry leaves. The RWCs of the red-light group for 3 days (RL1), blue-light group for 3 days (BL1), and 50% red-/50% blue-light group for 3 days (RBL1) were decreased compared with that of the no-light group for 3 days (NL1). And the 7-day treatments have results similar to those of the 3-day treatments (Table 1).

The content of the photosynthetic pigment is affected by the quality of the light [1]. In this study, the LL and NL treatments increased the contents of chlorophyll a (Chl a) and chlorophyll b (Chl b) compared with CK, respectively. RL1, BL1, and RBL1 decreased the contents of Chl a and Chl b compared with NL1. The red-light group for 7 days (RL2) and 50% red/50% blue-light group for 7 days (RBL2) increased the content of Chl a compared with the no-light group for 7 days (NL2). RBL2 increased the content of Chl b, while the blue-light group for 7 days (BL2) decreased the contents of Chl a and Chl b compared with NL2 (Table 1).

The photosynthetic rate (Pn) is an important indicator for detecting the intensity of the photosynthesis [19]. The minimal fluorescence (F_0_) is the fluorescence yield obtained when the reaction center of photosystem II is completely open; this value provides insight into the activity of the reaction center of PSII [20]. In this study, the LL and NL treatments decreased the Pn and F_0_ levels of the strawberry leaves compared with CK. The RL, BL, and RBL treatments increased the Pn and F_0_ levels of the strawberry leaves compared with NL treatment (Table 1).

### 2.2. Analysis of Phytohormones

The light quality affects the content of phytohormones in strawberry plants [13]. In this study, RL2, BL2, and RBL2 decreased the indole-3-acetic acid (IAA) content compared with NL2. LL1 and NL1 reduced the 3-indolebutyric acid (IBA) content compared with CK1 (Table 2).

NL1 increased the indole-3-carboxylic acid (ICA) content compared with CK1, and BL1 decreased the ICA content compared with NL1. LL2 and NL2 reduced the ICA content compared with CK2. RBL2 decreased the ICA content compared with NL2 (Table 2).

LL1 increased the indole-3-carboxaldehyde (I3A) content compared with CK1. BL1 decreased the I3A content when compared with NL1. RBL1 elevated the I3A content compared with NL1. RBL2 decreased the I3A content compared with NL2 (Table 2).

N6-isopentenyladenine (IP) was only detected in BL1 and BL2. RL1 and BL1 decreased the isopentenyl adenosine (IPA) content, and RBL1 increased the IPA content compared with NL1. LL2 reduced the IPA content compared with CK2. NL2 elevated the IPA content when compared with CK2. BL2 and RBL2 decreased the IPA content compared with NL2 (Table 2).

LL1 increased the trans-zeatin-riboside (*t*Z) content compared with CK1. NL2 elevated the trans-zeatin-riboside (*t*Z) content compared with CK2. The BL and RBL treatments decreased the *t*Z content separately compared with NL. There were no significant differences observed among the contents of kinetin (Table 2).

LL1 increased the methyljasmonate (MeJA) content, and NL1 decreased the MeJA content when compared with CK1. RL1, BL1, and RBL1 reduced the content of MeJA compared with NL1. And among the 7-day treatments, MeJA was only detected in CK2 (Table 2).

The LL and NL treatments decreased the N-jasmonic acid isoleucine (JA-Ile) content compared with CK. RL1, BL1, and RBL1 reduced the JA-Ile content compared with NL1 (Table 2).

RL1, BL1, and RBL1 decreased the jasmonic acid (JA) content compared with NL1. LL2 and NL2 reduced the content compared with CK2. RL2, BL2, and RBL2 increased the JA content compared with NL2 (Table 2).

LL1 and NL1 decreased the salicylic acid (SA) content compared with CK1. LL2 reduced the SA content compared with CK2. RL2, BL2, and RBL2 decreased the SA content compared with NL2 (Table 2).

The LL treatment decreased the abscisic acid (ABA) content compared with CK, RL2, and BL2; RBL2 increased the ABA content compared with NL (Table 2).

### 2.3. Analysis of Differential Genes in Transcriptome

In the five comparisons of the 3-day groups, 7572 DEGs were detected (q value = 0.01, log2 fold change = 3), and 104 common DEGs were identified: 1558 DEGs were only found in CK1 vs. NL1; 779 DEGs were only found in LL1 vs. NL1; 2139 DEGs were only found in RL1 vs. NL1; 126 DEGs were only found in BL1 vs. NL1; 402 DEGs were only found in RB1 vs. NL1 (Figure 1a).

In the five comparisons of the 7-day group that were performed, 9935 DEGs were observed (q value = 0.01, log2 fold change = 3), and 447 common DEGs were identified: 3276 DEGs were only found in CK2 vs. NL2; 1213 DEGs were only found in LL2 vs. NL2; 105 DEGs were only found in RL2 vs. NL2; 83 DEGs were only found in BL2 vs. NL2; 241 DEGs were only found in RBL2 vs. NL2 (Figure 1b).

A high proportion of DEGs was associated with molecular functions, including photosynthesis antenna proteins (69), photosynthesis (68), carbon fixation in photosynthetic organisms (80), porphyrin and chlorophyll metabolisms (37), carotenoid biosynthesis (35), tryptophan metabolism (29), phenylalanine metabolism (19), zeatin biosynthesis (13), and *alpha*-linolenic acid metabolism (46) (Figure 1c).

### 2.4. Analysis of Module–Trait Relationships

This result indicated that the content of IAA exhibited a significant positive correlation with the gene FPKM (fragments per kilobase of exon model per million mapped fragments) in the MEred and MEmagenta modules but a significant negative correlation with the gene FPKM in the MEpurple and MEorange modules. The content of *t*Z exhibited a significant negative correlation with the gene FPKM in the MElightyellow module. The content of JA exhibited a significant positive correlation with the gene FPKM in the MEskyblue module. The content of SA exhibited a significant positive correlation with the gene FPKM in the MEturquoise, MEred, MElightgreen, MEdarkred, and MEwhite modules but a significant negative correlation with the gene FPKM in the MEorange, MEyellow, and MEpurple modules. The content of ABA exhibited a significant positive correlation with the gene FPKM in the MEtan module. The content of RWC exhibited a significant positive correlation with the gene FPKM in the MEbrown module and a significant positive correlation with the gene FPKM in the MElightyellow module. The content of Chl a exhibited a significant negative correlation with the gene FPKM in the MEgreenyellow and MEdarkgreen modules. The content of Chl b exhibited a negative correlation with the gene FPKM in the MEgreenyellow and MEblue modules. The level of F_0_ exhibited a significant negative correlation with the gene FPKM in the MEgreen, MEbrown, MEblack, and MEmagenta modules. The level of the Pn exhibited a significant positive correlation with the gene FPKM in the MEturquoise, MEblue, and MEdarkgreen modules but a significant negative correlation with the gene FPKM in the MEyellow, MEbrown, MEgreen, MEsteelblue, and MEtan modules (Figure 2).

### 2.5. Analysis of 4 Photosynthesis-Related Pathways

#### 2.5.1. Chlorophyll Metabolism

Glutamyl-tRNA reductase 1 (HEM11) is the first enzyme involved in tRNA-dependent tetrapyrrole biosynthesis [21]. S-adenosyl-L-methionine-dependent uroporphyrinogen III methyltransferase (UPM1) catalyzes the SAM-dependent methylations of uroporphyrinogen III at positions C-2 and C-7 to form precorrin-2 via precorrin-1 [22]. Ferrochelatase (HemH) catalyzes the ferrous insertion into protoporphyrin IX [23]. The cytochrome c oxidase assembly protein COX15 (COX15) is required for the hydroxylation of heme O in the formation of heme A [24]. The magnesium-chelatase subunit ChlH (CHLH) catalyzes the insertion of magnesium ions into protoporphyrin IX to yield Mg-protoporphyrin IX in chlorophyll synthesis [25]. Magnesium-protoporphyrin IX monomethyl ester (oxidative) cyclase (CRD1) results in the formation of divinylprotochlorophyllide from Mg-protoporphyrin IX [26]. Divinyl chlorophyllide a 8-vinyl-reductase (DCVR) participates in the process for forming NADP^+^ + protochlorophyllide a = 3,8-divinyl protochlorophyllide a + H^+^ + NADPH [27]. Protochlorophyllide reductase (PORA) aids in the phototransformation of protochlorophyllide to chlorophyllide [28]. Chlorophyll synthase (CHLG) catalyzes the esterification of chlorophyllide a or b [29]. Chlorophyllase-1 (CLH1) catalyzes the hydrolysis of the ester bond in chlorophyll to yield chlorophyllide and phytol [30]. Magnesium dechelatase SGRL (SGRL) is involved in the degradation of chlorophylls a and b in chlorophyll–protein complexes [31]. Geranylgeranyl diphosphate reductase (CHLP) catalyzes the reduction of geranylgeranyl bacteriochlorophyllides a and b to bacteriochlorophylls a and b [32]. In this study, there were 37 DEGs found to be involved in the porphyrin and chlorophyll metabolisms; the expressions of three *HEM*, two *HENH2*, two *CHLH*, four *CRD1*, three *DCVR*, four *PORA*, one *CHLG*, three *SGRL*, two *CLH1*, and four *CHLP* genes exhibited a positive correlation with Pn, while the expression of *HEMH* exhibited a negative correlation with Pn based on the results of the WGCNA (Figure 3).

#### 2.5.2. Photosynthesis

Photosynthesis is closely related to the quality and intensity of the light [5]. In this study, there were 68 DEGs observed during photosynthesis; the expressions of three photosystem II protein protein P (*PsbP*), three photosynthetic NDH subunit of lumenal location 3 (*PsbQ*), one photosystem II 10 kDa polypeptide (*PsbR*), two photosystem II 22 kDa protein (*PsbS*), one photosystem II reaction center W protein *(PsbW*), five photosystem II core complex protein photosystem II protein Y (*PsbY*), three photosystem II repair protein PSB27-H1(*Psb27*), two photosystem II reaction center PSB28 protein (*Psb28*), four photosystem I reaction center subunit II (*PsaD*), three photosystem I reaction center subunit IV A (*PsaE*), two photosystem I reaction center subunit III (*PsaF*), three photosystem I reaction center subunit V (*PsaG*), two photosystem I reaction center subunit VI (*PsaH*), four photosystem I reaction center subunit K (*PsaK*), one photosystem I reaction center subunit XI (*PsaL*), one photosystem I reaction center subunit N (*PsaN*), four photosystem I subunit O (*PsaO*), two cytochrome b6-f complex iron–sulfur subunit (*PetC*), five ferredoxin–NADP reductase (*PetH*), and four ATP synthase delta chain (*Delta*) genes exhibited a positive correlation with Pn, while the expressions of three ferredoxin C 1 (*PetF*) genes exhibited a negative correlation with Pn based on the results of the WGCNA (Figure 4).

#### 2.5.3. Photosynthesis Antenna Proteins

The light-harvesting complex in photosystem I (Lhca) functions as a light receptor that captures and delivers excitation energy to photosystems [33]. The light-harvesting complex in photosystem II (Lhcb) modulates the rate of the state transition in photosystem II and influences its macrostructure; it is also involved in the transfer of PSII excitation energy and charge separation during photosynthesis in thylakoids [34]. There are four types of Lhca in strawberry leaves and seven types of Lhcb in strawberry leaves. In this study, 69 DEGs associated with photosynthesis antenna proteins were observed; the expressions of four *Lhca1*, six *Lhca2*, four *Lhca3*, three *Lhca4*, fourteen *Lhcb1*, four *Lhcb2*, three *Lhcb3*, eight *Lhcb4*, four *Lhcb5*, four *Lhcb6*, and one *Lhcb7* genes exhibited a positive correlation with Pn based on the results of the WGCNA (Figure 5).

#### 2.5.4. Carbon Fixation in Photosynthetic Organisms

Phosphoenolpyruvate carboxylase (CAPP) forms oxaloacetate through the carboxylation of phosphoenolpyruvate and CO_2_ (atmospheric) [35], and malate dehydrogenase (MDHP or MDHG) catalyzes the NAD/NADH-dependent interconversion of the substrates malate and oxaloacetate [36]. Aspartate aminotransferase (AATC) is a bidirectional enzyme that exists between oxaloacetate and aspartate. The NADP-dependent malic enzyme (MAOX) releases CO_2_ and pyruvate from malate. Glutamate glyoxylate aminotransferase 2 (GGT2) is a bidirectional enzyme that exists between pyruvate and L-alanine. Pyruvate phosphate dikinase (PPDK) is involved in the formation of phosphoenolpyruvate. The released CO_2_ can be fixed by D-ribulose 1,5-bisphosphate using H_2_O and the ribulose bisphosphate carboxylase small subunit (RBS), which forms glyceraldehyde 3P. Glyceraldehyde 3P can be converted to D-fructose 6P under the actions of the phosphoglycerate kinase (PGKH), glyceraldehyde-3-phosphate dehydrogenase B (G3PP1), fructose-bisphosphate aldolase 2 (ALFP2), and fructose-1,6-bisphosphatase (F16P2) enzymes. D-fructose 6P can be converted to D-ribulose 5P via two ways: The first path involves the conversion from D-fructose 6P to xylulose 5P and the conversion from xylulose 5P to D-ribulose 5P under the catalysis of ribulose-phosphate 3-epimerase (RPE). In the second pathway, D-fructose 6P is converted to D-erythrose 4P, and D-erythrose 4P and glycerone phosphate are converted to sedoheptulose 1,7-bisphosphate under the catalysis of fructose-bisphosphate aldolase 2 (ALFP2); then, sedoheptulose 1,7-bisphosphate is converted to D-ribulose 5P under the actions of sedoheptulose-1,7-bisphosphatase (S17P) and ribose-5-phosphate isomerase (RPI), and D-ribulose 5P is converted to D-Ribulose 1,5-bisphosphate under the action of phosphoribulokinase (KPPR). In this study, we drew the terms for the enriched pathway of the carbon fixation in photosynthetic organisms and found 80 DEGs. The expressions of one *MDHP*, four *MDHG*, one *MAOX*, two *PCKA*, two *GGT2*, three *PPDK*, four *RBS*, six *PGK*, one *G3PP1*, four *G3PB*, three *G3PA2*, four *ALFP2*, two *F16P2*, one *F16P1*, one *TPIS*, one *TIM*, five *RPI3*, five *KPPR*, and two *PRE* genes exhibited a positive correlation with Pn, while the expression of the *S17P* gene showed a negative correlation with Pn based on the results of the WGCNA (Figure 6).

### 2.6. Analysis of Hormone Synthesis and Signal Transduction

#### 2.6.1. ABA Synthesis and Signal Transduction

Phytoene synthase (PSY) catalyzes the reaction from geranylgeranyl pyrophosphate (GGPP) to phytoene, lycopene epsilon cyclase (LCYE) catalyzes the double cyclization reaction that converts lycopene to βcarotene, and β-carotene hydroxylase 2 (BCH2) is involved in the biosynthesis of zeaxanthin and lutein. Carotenoid cleavage dioxygenase (CCD) may be involved in the cleavage of carotenoids, and xanthoxin dehydrogenase (ABA2) is involved in the biosynthesis of ABA and *s*-violaxanthin; meanwhile, violaxanthin de-epoxidase (VDE) is involved in the violaxanthin cycle that controls the concentration of zeaxanthin. Also, 9-*cis*-epoxycarotenoid dioxygenase NCED catalyzes the biosynthesis of xanthoxin from 9-*cis*-violaxanthin and 9’-*cis*-neoxanthin, and abscisic acid 8’-hydroxylase (ABAH) is involved in the oxidative degradation of ABA; meanwhile, the ABA receptor PYL can be activated via a change in ABA and can inhibit the activity of group-A protein phosphatases type 2C (PP2Cs) when activated by ABA. PP2Cs are a key component and repressor of the ABA signaling pathway, which regulates stomatal closure, seed germination, and the inhibition of vegetative growth and resistance. Serine/threonine-protein kinase (SAPK) may play a role in the signal transduction of hyperosmotic responses. In this study, there were 70 DEGs associated with ABA synthesis and signal transduction; the expressions of two *BCH2*, one *CCD4*, one *VDE*, one *PYL5*, nine *P2Cs*, and one *SAPK1* genes exhibited a positive correlation with Pn, while the expressions of two *PYL4*, one *PYL11*, and two *SAPK2* genes exhibited a negative correlation with Pn based on the results of the WGCNA (Figure 7).

#### 2.6.2. IAA Synthesis and Signal Transduction

There are three routes by which L-tryptophan (Trp) can be converted to IAA. In the first pathway, tryptophan aminotransferase-related protein 2 (TAR2) converts Trp to indole-3-pyruvic acid, which is then converted to IAA under the actions of indole-3-pyruvate monooxygenases YUC 3, YUC 5, and YUC 10. In the second pathway, Trp can be converted to indole-3-thiohydroximate and then converted to indolylmethyl-desulfoglucosinolate by UDP-glycosyltransferase 74B1; it is then finally converted to IAA. In the third pathway, Trp can be converted to indole-3-acetamide and then converted to IAA by amidase (AMI). IAA can be converted to inactive IAA under the actions of benzaldehyde dehydrogenase (BALDH), aldehyde dehydrogenase family 3 member F (AL3F), and 2-oxoglutarate-dependent dioxygenase (DAO). The auxin-transporter-like protein (LAX), auxin-induced protein (AUX), auxin-responsive protein (IAA), auxin-responsive protein SAUR50, indole-3-acetic acid-amido synthetase GH31, and indole-3-acetic acid-induced protein ARG7 are downstream regulatory genes of IAA. In this study, there were 70 DEGs found to be associated with auxin (IAA) synthesis and signal transduction; the expressions of one *AL2B4*, four *AL3I1*, one *AMI4G*, and one *GH310* genes exhibited a positive correlation with Pn, while the expressions of two *AL2B7*, one *AX22D*, and five auxin-induced protein genes exhibited a negative correlation with Pn based on the results of the WGCNA (Figure 8).

#### 2.6.3. *t*Z Synthesis and Signal Transduction

Adenylate isopentenyltransferase 3 (IPT3) catalyzes the transfer of an isopentenyl group from dimethylallyl diphosphate (DMAPP) to ATP, ADP, or AMP. Cytokinin dehydrogenase 3 catalyzes the oxidation of cytokinins, and UDP-glycosyltransferase 73C belongs to the UDP-glycosyltransferase family. Histidine kinase 2 (AHK2) feeds phosphates to the phosphorelay-integrating histidine phosphotransfer protein and activates a subsequent cascade in the presence of cytokinins. The histidine-containing phosphotransfer protein (AHP) functions as a two-component phosphorelay mediator between cytokinin sensor histidine kinases and response regulators (B-type ARRs), while the two-component response regulator ORR acts as a negative regulator of cytokinin signaling; in addition, the two-component response regulator ARR can directly activate some type-A response regulators in response to cytokinins. In this study, there were 38 DEGs found to be associated with cytokinin synthesis and signal transduction; the expression of the *ORR23* gene exhibited a positive correlation with Pn, while the expressions of three *CKX7*, two *PHP5*, and three *ARR4* genes exhibited a negative correlation with Pn based on the results of the WGCNA (Figure 9).

#### 2.6.4. Jasmonic Acid (JA) Synthesis and Signal Transduction

Phospholipase A2-alpha (PLA2A) releases lysophospholipids (LPLs) and free fatty acids (FFAs) from membrane phospholipids [37]. Linoleate 13S-lipoxygenase (LOX) is involved in the production of 13(S)-hydroperoxy-9,11-octadecadienoic acid [38]. Allene oxide cyclase (AOC) is involved in the production of 12-oxo-phytodienoic acid (OPDA), while 12-oxophytodienoate reductase (OPR) reduces 12-OPDA to OPC-8:0. OPC8-CoA is converted to JA with the participations of acyl-coenzyme A oxidase 2 (ACOX2) and 3-ketoacyl-CoA thiolase 2 (THIK2). Jasmonic acid carboxyl methyltransferase 2 (JMT2) catalyzes the methylation from jasmonate to methyljasmonate. The transcription factor MYC2 additively controls the subsets of JA-dependent responses and can be induced by light and the JA signaling pathways. The protein TIFY 10A interacts with COI1 and inositol pentakisphosphate to form a high-affinity jasmonate coreceptor. In this study, there were 55 DEGs found to be associated with JA synthesis and signal transduction; the expressions of three *LOX21*, two *HPL1*, one *QORH*, and two *ACO* genes exhibited a positive correlation with Pn, while the expressions of two *ACOX2* genes exhibited a negative correlation with Pn based on the results of the WGCNA (Figure 10).

#### 2.6.5. Salicylic Acid (SA) Synthesis and Signal Transduction

Phenylpyruvate and phenylalanine can reciprocally transform under the actions of bifunctional aspartate aminotransferase, glutamate/aspartate–prephenate aminotransferase (AATC/PAT), and aminotransferase TAT. Phenylalanine ammonia-lyase 1 (PAL) catalyzes the reaction that is involved in the biosynthesis of trans-cinnamate from L-phenylalanine. Isochorismate synthase 2 (ICS2) participates in the conversion from chorismate to isochorismate [39]. Salicylic-acid-binding protein 2 (SABP2) is required to convert methyl salicylate (MeSA) to SA. The BTB/POZ domain and ankyrin repeat-containing protein NPR1 are activators of the systemic acquired resistance [40], while the transcription factor TGA7 mediates IAA- and SA-inducible transcriptions. Pathogenesis-related protein 1 (PRB1) is considered to be an important protein that defends against abiotic stress and disease [41]. In this study, there were 30 DEGs found to be associated with SA synthesis and signal transduction; the expressions of three *PAT*, one *TAT2*, two *PAL1*, and one *NPR1* genes exhibited a positive correlation with Pn, while the expressions of two *TGA4* genes exhibited a negative correlation with Pn based on the results of the WGCNA (Figure 11).

### 2.7. PPI Analysis

We analyzed the protein interaction relationships that were observed in 103 proteins corresponding to 258 DEGs. We then drew a diagram representing the core protein interactions associated with light regulation according to PPI scores above 0.70; this culminated in 46 proteins. There were 59 proteins corresponding to 122 DEGs in CK1 vs. NL1, and there were 83 proteins corresponding to 185 DEGs in CK2 vs. NL2. There were 21 proteins corresponding to 74 DEGs in LL1 vs. NL1, and there were 58 proteins corresponding to 203 DEGs in LL2 vs. NL2. There were 15 proteins corresponding to 35 DEGs in RL1 vs. NL1, and there were 47 proteins corresponding to 118 DEGs in RL2 vs. NL2. There were 7 proteins corresponding to 11 DEGs in BL1 vs. NL1, and there were 47 proteins corresponding to 87 DEGs in BL2 vs. NL2. There were 6 proteins corresponding to 9 DEGs in RBL1 vs. NL1, and there were 43 proteins corresponding to 79 DEGs in RBL2 vs. NL2 (Figure 12 and Appendix A).

## 3. Discussion

### 3.1. Effects of Different Light Qualities on Chlorophyll Metabolism

In this study, we found that the LL1 and NL1 treatments increased the Chl a and Chl b contents, respectively, compared with CK1, while RL1, BL1, and RBL1 decreased the Chl a and Chl b contents compared with NL1 (Table 1). This is consistent with the finding that the photosynthetic pigment content increased under low-light conditions [1]. Furthermore, we screened 37 DEGs associated with chlorophyll metabolism via WGCNA, and the PPI results showed that HEM11, CHLH, CRD1, PORA, SGRL, and CHLP participate in light regulation in strawberry leaves (Figure 12 and Appendix A). Previous studies have shown that HEM11 belongs to the glutamyl-tRNA reductase family and regulates tRNA-dependent tetrapyrrole biosynthesis [21]. In addition, CHLH is correlated with plastid signaling [25]; CRD1 catalyzes the formation of the isocyclic ring in chlorophyll biosynthesis [26]; PORA catalyzes the reduction from protochlorophyllide to chlorophyllide [28]; SGRL regulates the chlorophyll metabolism and contributes to normal plant growth and development [31]; CHLP actively mediates the growth and development of poplar by regulating photosynthesis [32]. From these data, we speculate that the quality of the light affects the chlorophyll metabolism through *HEM11*, *CHLH*, *CRD1*, *PORA*, *SGRL*, and *CHLP* genes and finally affects the pigment content.

### 3.2. Effects of Different Light Qualities on Photosynthesis

In this study, we found that the LL and NL treatments decreased the Pn and F_0_ of the strawberry leaves, respectively, compared with the CK treatment. The RL, BL, and RBL treatments increased the Pn and F_0_ of the strawberry leaves compared with the NL treatment (Table 1). Furthermore, we screened 68 DEGs associated with photosynthesis via WGCNA, and the PPI results showed that the PsaD, PsaE, PsaF, PsaK, PsaL, PsaO, PetH, Psb27, PsbR, PsbW, and delta proteins participate in light regulation in strawberry leaves (Figure 12 and Appendix A). Previous studies have shown that PsaD is located on the stromal side of the thylakoids and that it assembles in vitro into the membranes in its precursor (pre-PsaD) and also in its mature (PsaD) form [42]. In addition, PsaE plays a role in O_2_ reduction in photosystem I and protects photosystem II against photoinhibition [43]; PsaF is involved in the docking of the electron donor proteins plastocyanin and cytochrome c_6_ in eukaryotic photosynthetic organisms [44]; PsaK has two transmembrane alpha-helices [45]; PsaL functions as a light-driven plastocyanin–ferredoxin oxidoreductase in the thylakoid membranes of cyanobacteria and higher plants [46]; PsaO is involved in the balance of the excitation energy between photosystems I and II; PetH controls the rate of the interaction with PetF [47]; Psb27 enables the quenching of excess light energy during its participation in the PSII lifecycle [48]; PsbR is a missing link in the assembly of the oxygen-evolving complex of PSII [49]; the PsbW protein stabilizes the supramolecular organization of photosystem II [50]; F-type ATPase delta produces ATP in cells. We found that the quality of the light affects the gene expressions of the PsaD, PsaE, PsaF, PsaK, PsaL, PsaO, PetH, Psb27, PsbR, PsbW, and delta proteins and finally impacts F_0_ and Pn.

### 3.3. Effects of Different Light Qualities on Photosynthesis Antenna Proteins

Photosystems comprise a reaction center and a peripheral antenna system, the latter of which is composed of Lhca and Lhcb proteins that offer relevant physiological functions in both light harvesting and photoprotection [51]. In this study, we screened 69 DEGs associated with photosynthesis via WGCNA, and the PPI results showed that the Lhca1, Lhca2, Lhca3, Lhca6, Lhcb1, Lhcb3, Lhcb4, Lhcb5, and Lhcb7 proteins participate in light regulation in strawberry leaves (Figure 12 and Appendix A). Previous studies have shown that Lhca1 is the main point at which the energy harvested by the antenna is delivered to the core, and it has a particular energy landscape with an unusual configuration of low-energy states compared to Lhca4 [52]. Lhca1/4 and Lhca2/3 form two red-emitting heterodimers in higher-plants’ photosystem I [53]. An Lhca1–Lhca4–Lhca2–Lhca3 belt attaches on one side, and an Lhca5–Lhca6 heterodimer associates on the other side between PsaG and PsaH [54]. LHCII comprises Lhcb1-2-3, where the Lhcb1 and Lhcb2 complexes have similar, but not identical, pigment-binding properties; meanwhile, Lhcb3 is clearly different with respect to both its pigment-binding and spectral properties and cannot produce homotrimers in vitro [55]. Lhcb4 is an ancestral LHCII that has three types, and the pivotal position of Lhcb4 in the PSII–LHCII supercomplex allows it to function as a linker that conjoins either the S- or M-trimers of LHCII to the PSII core [56]. The responses of Lhcb5 to abiotic stress and Lhcb5 knock-out results in a thylakoid arrangement, grana membranes, and a starch granule [57]. Lhcb7 shifts the response of leaves to irradiance with regard to their photosynthetic light-harvesting regulation [58]. These results imply that the quality of the light affects the gene expressions of *Lhca1*, *Lhca2*, *Lhca3*, *Lhca6*, *Lhcb1*, *Lhcb3*, *Lhcb4*, *Lhcb5*, and *Lhcb7* in terms of photosynthesis antenna proteins.

### 3.4. Effects of Different Light Qualities on Carbon Fixation

Light modulates the biosynthesis and organization of the carbon fixation machinery via the flow of photosynthetic electrons [59]. In this study, we found that the LL and NL treatments increased the RWC in strawberry leaves compared with the CK treatment. The RL, BL, and RBL treatments decreased the RWC in strawberry leaves compared with the NL treatment (Table 1). Furthermore, we screened 80 DEGs associated with photosynthesis via WGCNA, and the PPI results showed that the MDHG, MAOX, PPDK, RBS, PGK, G3PB, G3PA2, RPI3, KPPR, and ALFP2 proteins participate in light regulation in strawberry leaves (Figure 12 and Appendix A). Previous studies have shown that MDHG catalyzes the NAD/NADH-dependent interconversion of the substrates malate and oxaloacetate [36]. MAOX releases CO_2_ and pyruvate from malate and participates in the response of the ABA [60]. PPDK enhances photosynthesis, carbon assimilation, and abiotic stress tolerance under conditions with elevated CO_2_ [61]. RBS catalyzes the rate-limiting step of CO_2_ fixation in photosynthesis [62]. PGK is a metabolic enzyme involved in glycolysis and the carbon reduction cycle [63]. G3PA2 and G3PB are essential for maintaining photosynthetic efficiency [64]. RPI3 is involved in the key activity of the pentose phosphate pathway [65]. KPPR is an essential enzyme of the CBB cycle in photosynthesis and catalyzes the ATP-dependent conversion from ribulose-5-phosphate to ribulose-1,5-bisphosphate [66]. ALFP2 is a highly conserved enzyme that is involved in glycolysis and gluconeogenesis. Therefore, we surmise that the quality of the light affects the gene expressions of *MDHG*, *MAOX*, *PPDK*, *RBS*, *PGK*, *G3PB*, *G3PA2*, *RPI3*, *KPPR*, and *ALFP2* in carbon fixation and decreases the RWC in strawberry leaves.

### 3.5. Effects of Different Light Qualities on Phytohormone Synthesis and Signal Transduction

Light mediates the hormonal regulations of plant growth and development [67]. In this study, we found that the RL2, BL2, and RBL2 treatments decreased the IAA content in strawberry leaves compared with NL2. The BL and RBL treatments decreased the *t*Z content compared with the NL treatment. The RL, BL, and RBL treatments decreased the SA content compared with the NL treatment. The BL and RBL treatments decreased the *t*Z content separately compared with NL. The LL treatment decreased the ABA content compared with CK; RL2, BL2, and RBL2 increased the ABA content compared with NL.

We screened 142 DEGs associated with the synthesis of phytohormones and signal transduction via WGCNA, and the PPI results showed that PYL5, P2C16, P2C37, P2C51, IAA4, AL3F1, AL3I1, ACOX2, PAT, and PAL1 participate in light regulation in strawberry leaves (Figure 12 and Appendix A). Previous studies have shown that the overexpression of *PYL*5 in rice inhibits growth and modulates gene expressions [68]. PP2Cs are negative regulators of ABA signaling and the adaptation of plants to stress [69]. IAA4 may mediate cell-specific responses to auxins [70]. IAA can be converted to indole-3-acetaldehyde under the actions of AL3F1 and AL3I1. ACOX2 contributes to JA production [71]. PAT contributes to the conversion from phenylpyruvate to L-phenylalanine, and PAL1 modulates the synthesis of SA from phenylalanine via cinnamic acid [39]. These results suggested that the light quality affects the gene expressions of *PYL5*, *P2C16*, *P2C37*, *P2C51*, *IAA4*, *AL3F1*, *AL3I1*, *ACOX2*, *PAT*, and *PAL1*; and the light quality finally affects the synthesis of phytohormones and signal transduction.

## 4. Materials and Methods

### 4.1. Plant Treatment and Sampling

‘Benihoppe’ strawberries were grown in greenhouses with sunlight at the Fruit Research Institute, Chinese Academy of Agricultural Sciences. We collected strawberry leaves at two time periods during the treatment: day 3 and day 7.

Two hundred strawberries in the control group (CK) were grown normally by absorbing solar energy (>1000 μmol·m^−2^·s^−1^), CK1 for 3 days and CK2 for 7 days. Two hundred strawberries in the low-intensity light group (LL) grew normally by absorbing solar energy but were covered with a 50% sunshade net in the greenhouse, LL1 for 3 days and LL2 for 7 days. The CK and LL groups were administered in the same greenhouse (23 ± 2 °C).

Two hundred strawberries in the no-light group were kept in an environment that simulated the no-light (NL) conditions, NL1 for 3 days and NL2 for 7 days. Two hundred strawberries were exposed to red light (RL) (80 μmol·m^−2^·s^−1^), RL1 for 3 days and RL2 for 7 days. Two hundred strawberries were exposed to blue light (BL) (80 μmol·m^−2^·s^−1^), BL1 for 3 days and BL2 for 7 days. Two hundred strawberries were exposed to red/blue light (RBL) (80 μmol·m^−2^·s^−1^) [5], RBL1 for 3 days and RBL2 for 7 days. The NL, RL, BL, and RBL groups were administered in the same greenhouse (4 ± 1 °C) but required it to be covered with a quilt.

### 4.2. Measurement of Relative Water Content

We collected 36 strawberry leaves and measured the fresh weight (FW), about 3 g; these leaves were taken back to the laboratory and dried in an oven at 80 °C for 48 h, and the dry weight (DW) was measured. We then calculated the relative water content (RWC) for three biological replicates as follows [72]:RWC = (FW − DW)/FW × 100%(1)

### 4.3. Measurement of Chl a and Chl b Contents

We collected 10 fourth leaves and drilled holes in the middle part of each leaf. Then, we weighed 0.1 g of the round-hole leaf tissue and put it into a 25 mL glass tube, added 20 mL of 96% anhydrous ethanol, and soaked the chlorophyll at room temperature in the dark for 3 days. During this period, we shook the glass tubes twice every day to fully extract the chlorophyll; this was repeated three times. The values at 665 nm and 649 nm were detected using a spectrometer for three biological replicates.

Then, we calculated the contents of Chl a and Chl b as follows [72]:Chl a (mg/L) = 13.95A_665_ − 6.88A_649_(2)
Chl b (mg/L) = 24.96A_649_ − 7.32A_665_(3)

### 4.4. Measurement of Pn and F_0_

We measured the Pn of the fourth leaves of 12 samples on the 3rd and 7th days using a Li-6400 portable photosynthesis system, and then we measured the F_0_ of the fourth leaves on the 3rd and 7th days using a chlorophyll fluorescence imaging system; the chlorophyll a fluorescence was examined on the adaxial (upper) surfaces of the leaves [20] for three biological replicates.

### 4.5. Measurement of Phytohormones

We collected 36 strawberry leaves that weighed 3 g each; these were stored in liquid nitrogen in a foam box surrounded by dry ice and transported to Novogene Co., Ltd. (Beijing, China). A series of phytohormone calibrators was made by mixing 24 individual phytohormones and then injected into an LC–MS system to develop standard curves. Next, 100 mg of strawberry leaves was homogenized with liquid nitrogen, homogenized with 400 μL of acetonitrile (50%) containing mixed internal standards, extracted at 4 °C for 30 min, and then centrifuged at 12,000 rpm for 10 min. The supernatant passed through the HLB sorbent and was subsequently eluted with 500 μL of acetonitrile (30%). These two fractions were collected in the same centrifuge tube and injected into the LC–MS system (ExionLC™AD UHPLC-QTRAP 6500+, AB SCIEX Corp., Boston, MA, USA) for the analysis of three biological replicates [73].

### 4.6. Measurement of Transcriptomic Reprogramming

The integrity of the total RNA of the strawberry leaves was assessed using an RNA Nano 6000 assay kit and a Bioanalyzer 2100 system (Agilent Technologies, Santa Clara, CA, USA). A library of strawberry leaves was prepared according to the method proposed by Novogene Co., Ltd., China and the library quality of the strawberry leaves was assessed using the Agilent Bioanalyzer 2100 system (Agilent Technologies, Santa Clara, CA, USA). The library preparations of the strawberry leaves were sequenced on an Illumina Novaseq platform by referring to the age of the strawberry [73] for three biological replicates.

### 4.7. Maps of Metabolic, Module–Trait Relationship, and Protein–Protein Interaction Networks

Metabolic maps were created using Photoshop 5 and Hemi 1.0 based on the FPKM values. The module–trait relationship map was created according to the WGCNA based on the values of the indices and the FPKM [74]. The protein–protein interaction network maps were created using Cytoscape_v3.8.1 by referring to the data of the strawberry [73].

### 4.8. Statistical Analysis

SPSS 20.0 software (IBM, Chicago, IL, USA) was used to analyze the one-way variances. All the data were the means of triplicate trials, and the standard deviation (SD) was calculated. Statistical differences in tables are referred to as significant for * *p* ≤ 0.05.

## Figures and Tables

**Figure 1 ijms-25-02765-f001:**
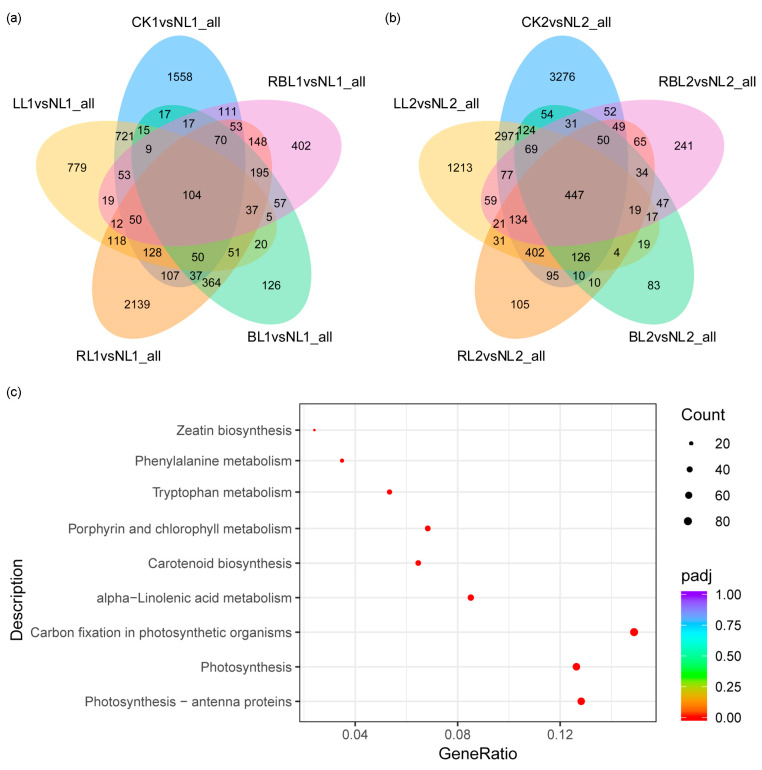
Analysis of the transcriptome in strawberry leaves under different light treatments. (**a**) Venn diagrams of differentially expressed genes in the five comparisons of the 3-day group. (**b**) Venn diagrams of differentially expressed genes in the five comparisons of the 7-day group. (**c**) The most enriched pathway terms related to the light quality.

**Figure 2 ijms-25-02765-f002:**
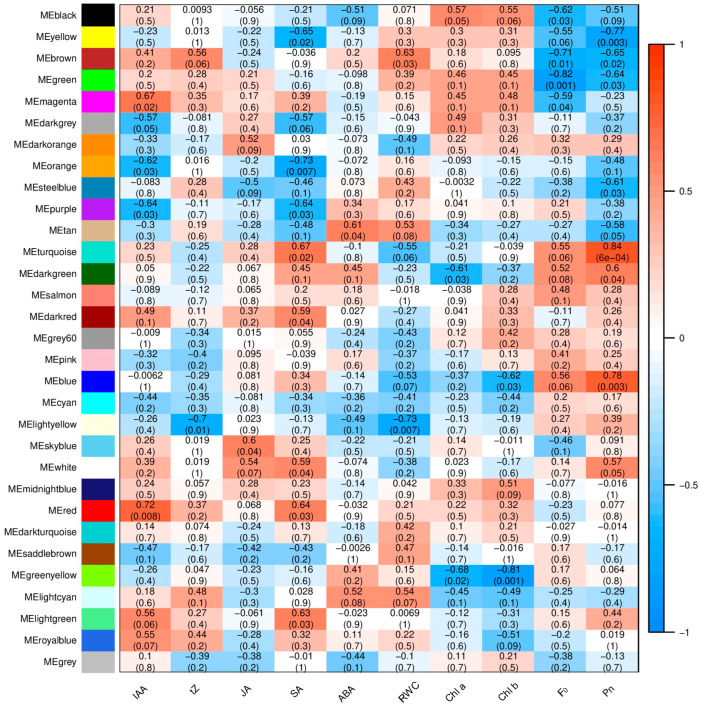
Module–trait relationships. The colors from blue to red are based on the value of the correlation between the modules (from −1 to 1); *p*-value < 0.05 indicates a significant correlation.

**Figure 3 ijms-25-02765-f003:**
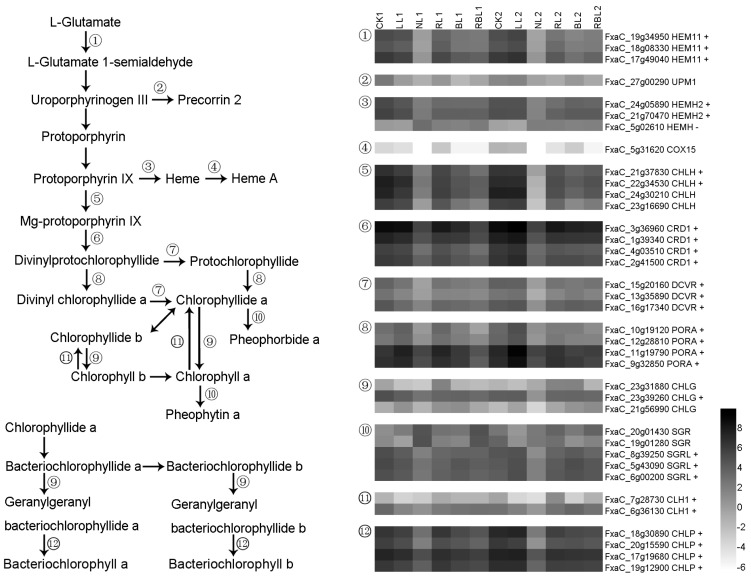
Heatmap of representative genes in chlorophyll metabolism. + indicates a positive correlation with Pn; − indicates a negative correlation with Pn; figure is in black and white.

**Figure 4 ijms-25-02765-f004:**
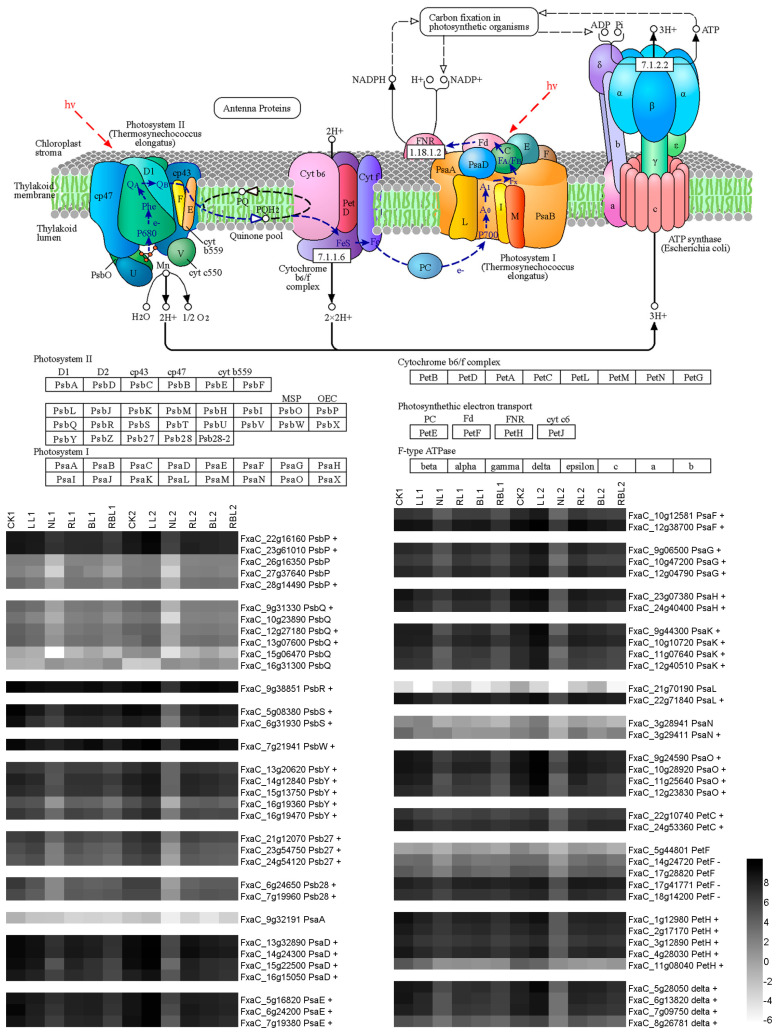
Heatmap of representative genes in photosynthesis. + indicates a positive correlation with Pn; − indicates a negative correlation with Pn; figure is in black and white.

**Figure 5 ijms-25-02765-f005:**
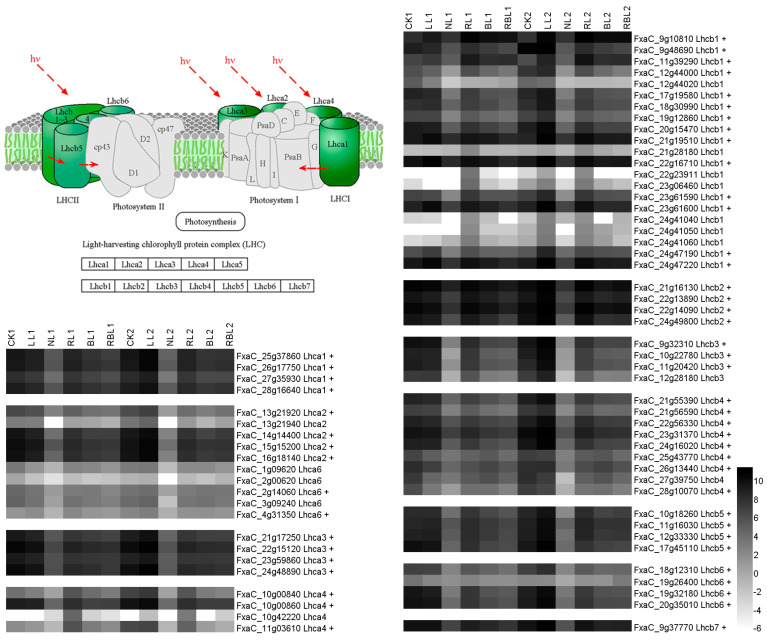
Heatmap of representative genes in antenna proteins. + indicates a positive correlation with Pn; − indicates a negative correlation with Pn; figure is in black and white.

**Figure 6 ijms-25-02765-f006:**
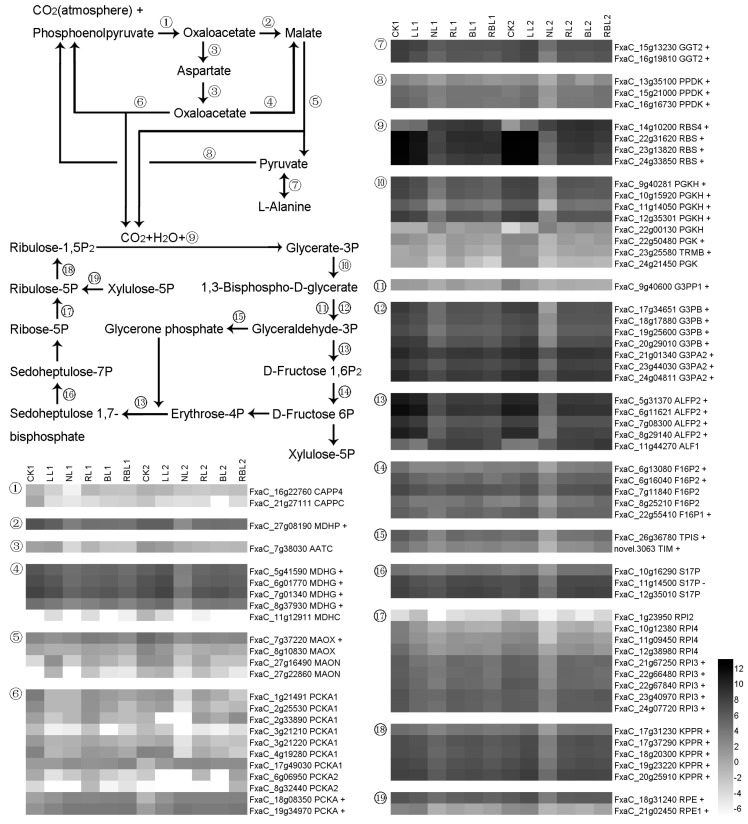
Heatmap of representative genes associated with carbon fixation in photosynthetic organisms. + indicates a positive correlation with Pn; − indicates a negative correlation with Pn; figure is in black and white.

**Figure 7 ijms-25-02765-f007:**
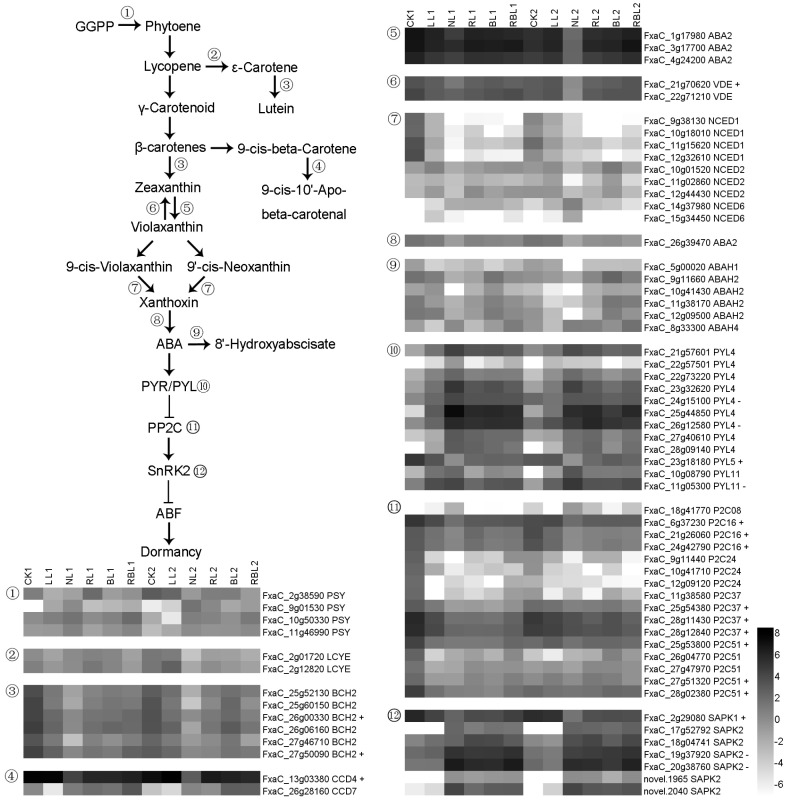
Heatmap of representative genes associated with ABA synthesis and signal transduction. + indicates a positive correlation with Pn; − indicates a negative correlation with Pn; figure is in black and white.

**Figure 8 ijms-25-02765-f008:**
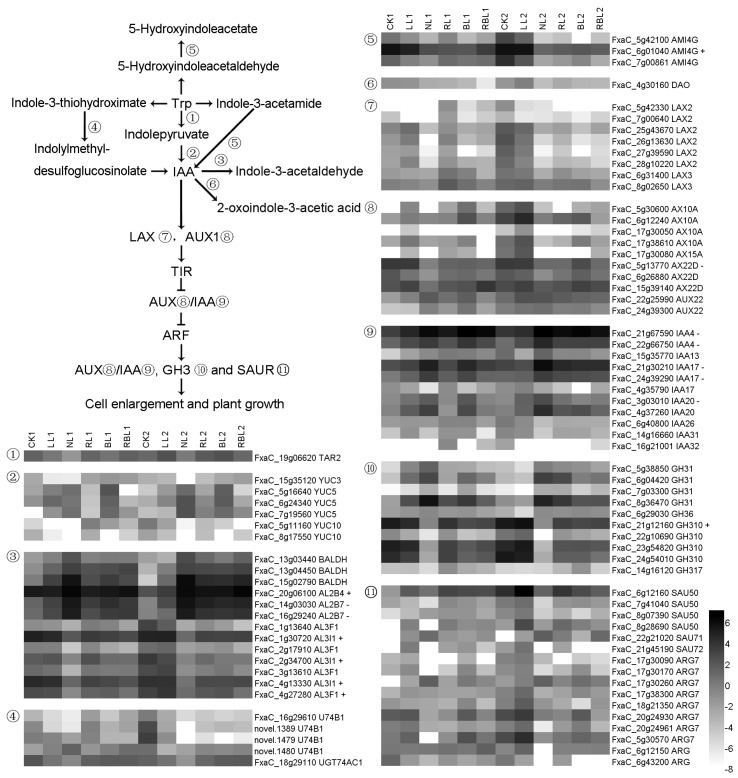
Heatmap of representative genes associated with IAA synthesis and signal transduction. + indicates a positive correlation with Pn; − indicates a negative correlation with Pn; figure is in black and white.

**Figure 9 ijms-25-02765-f009:**
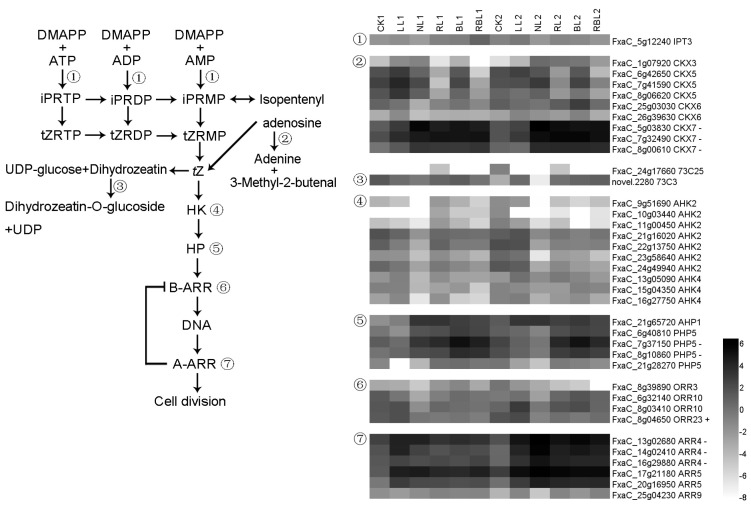
Heatmap of representative genes associated with *t*Z synthesis and signal transduction. + indicates a positive correlation with Pn; − indicates a negative correlation with Pn; figure is in black and white.

**Figure 10 ijms-25-02765-f010:**
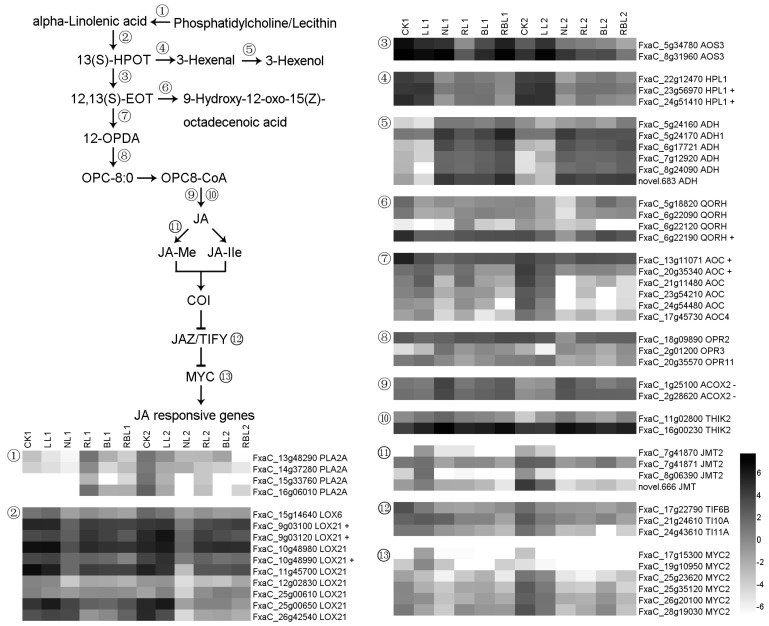
Heatmap of representative genes associated with JA synthesis and signal transduction. + indicates a positive correlation with Pn; − indicates a negative correlation with Pn; figure is in black and white.

**Figure 11 ijms-25-02765-f011:**
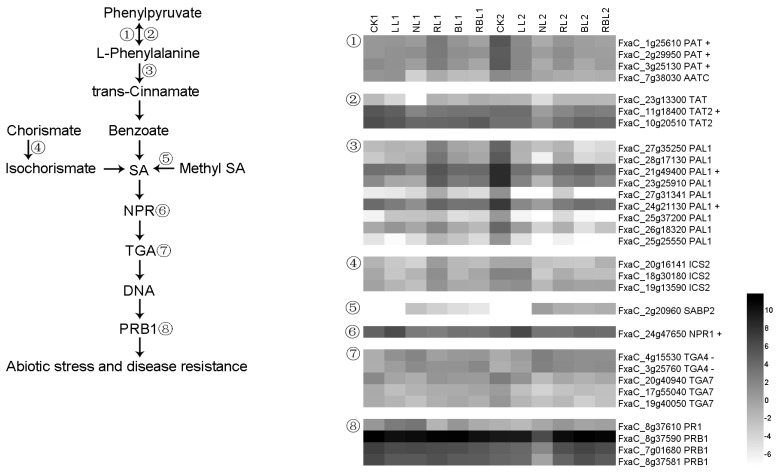
Heatmap of representative genes associated with SA synthesis and signal transduction. + indicates a positive correlation with Pn; − indicates a negative correlation with Pn; figure is in black and white.

**Figure 12 ijms-25-02765-f012:**
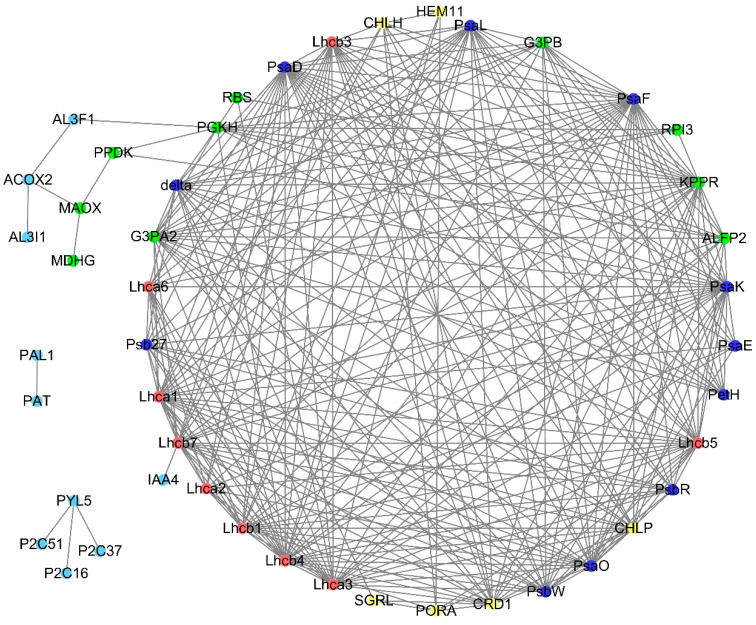
Core protein interaction diagram of light regulation in strawberry leaves. There are 10 light-blue nodes associated with hormone synthesis and signal transduction, 11 dark-blue nodes associated with photosynthesis, 6 yellow nodes associated with chlorophyll metabolism, 9 red nodes associated with photosynthesis antenna proteins, and 10 green nodes associated with carbon fixation in photosynthetic organisms. PPI scores show high confidence above 0.70.

**Table 1 ijms-25-02765-t001:** RWC, Chl a, Chl b, Pn, and F_0_ of strawberry leaves.

	CK1	LL1	NL1	RL1	BL1	RBL1	CK2	LL2	NL2	RL2	BL2	RBL2
RWC (%, FW)	67.51 ± 1.27 F	71.23 ± 1.12 DE	71.48 ± 0.39 CDE	69.85 ± 0.08 E	70.29 ± 0.37 E	70.57 ± 0.02 E	67.1 ± 0.11 F	73.38 ± 0.06 AB	74.57 ± 0.66 A	73.04 ± 1.13 ABC	72.93 ± 0.65 BC	72.64 ± 0.01 BCD
Chl a (mg/g, FW)	1.35 ± 0.2 AB	1.60 ± 0.06 AB	1.70 ± 0.06 A	1.37 ± 0.21 AB	1.52 ± 0.11 AB	1.61 ± 0.04 AB	1.34 ± 0.13 AB	1.45 ± 0.23 AB	1.43 ± 0.31 AB	1.47 ± 0.38 AB	1.11 ± 0.24 BC	1.82 ± 0.08 A
Chl b (mg/g, FW)	0.44 ± 0.05 AB	0.53 ± 0.02 A	0.54 ± 0.02 A	0.42 ± 0.07 ABC	0.44 ± 0.03 AB	0.50 ± 0.01 AB	0.15 ± 0.04 D	0.42 ± 0.07 ABC	0.35 ± 0.09 BC	0.35 ± 0.11 BC	0.28 ± 0.08 CD	0.47 ± 0.02 AB
Pn (μmol/(m^2^·s))	27.73 ± 0.4 A	8.30 ± 0.2 D	−3.60 ± 0.1 I	1.90 ± 0.1 E	0.63 ± 0.06 H	0.87 ± 0.06 G	32.47 ± 1.16 B	10.00 ± 0.1 C	−3.10 ± 0.1 I	2.43 ± 0.06 E	1.00 ± 0.1 FG	1.77 ± 0.06 EF
F_0_	3790 ± 385 AB	3536 ± 150 AB	2328 ± 257 C	3305 ± 349 B	2629 ± 362 C	3533 ± 181 AB	4165 ± 127 A	3673 ± 316 AB	2546 ± 222 C	3637 ± 484 AB	3423 ± 176 AB	3988 ± 253 AB

Notes: Mean ± SD; different capital letters indicate significant differences between treatments or stages according to one-way ANOVA at *p* < 0.01. FW, fresh weight.

**Table 2 ijms-25-02765-t002:** Phytohormone levels in strawberry leaves.

	CK1	LL1	NL1	RL1	BL1	RBL1	CK2	LL2	NL2	RL2	BL2	RBL2
Indole-3-acetic acid	6.72 ± 1.21 AB	6.43 ± 0.68 ABC	5.07 ± 2.65 BCD	2.02 ± 0.56 D	5.50 ± 1.33 ABCD	5.06 ± 1 BCD	6.23 ± 1.76 ABC	3.73 ± 0.83 BCD	8.73 ± 2.32 A	3.80 ± 0.6 BCD	3.33 ± 0.73 BCD	3.14 ± 0.36 CD
3-Indolebutyric acid	67.57 ± 1.48 A	34.00 ± 2.27 C	40.92 ± 13.85 BC	43.87 ± 4.88 BC	40.21 ± 5.49 BC	56.76 ± 12.26 AB	53.42 ± 14.2 ABC	48.01 ± 3.23 ABC	43.46 ± 7.07 BC	46.40 ± 8.94 BC	39.33 ± 8.08 BC	37.89 ± 3.24 BC
Indole-3-carboxylic acid	2.89 ± 0.93 BCDEF	4.57 ± 0.66 AB	5.61 ± 1.54 A	3.99 ± 0.57 ABCD	1.06 ± 0.46 EF	3.76 ± 0.48 ABCD	0.77 ± 0.67 F	3.30 ± 0.81 ABCDE	4.19 ± 0.89 ABC	3.19 ± 1.85 ABCDE	1.79 ± 0.34 CDEF	1.71 ± 0.98 DEF
Indole-3-carboxaldehyde	5.18 ± 1.86 CD	14.53 ± 1.03 A	5.17 ± 2.67 CD	5.61 ± 0.9 C	1.29 ± 0.5 E	10.95 ± 1.33 B	4.03 ± 2.06 CDE	7.32 ± 0.77 C	6.73 ± 1.12 C	6.17 ± 1.44 C	5.03 ± 1.6 CD	1.69 ± 1.14 DE
N6-Isopentenyladenine	0 ± 0	0 ± 0	0 ± 0	0 ± 0	0.48 ± 0.03 A	0 ± 0	0 ± 0	0 ± 0	0 ± 0	0 ± 0	0.31 ± 0.06 B	0 ± 0
Isopentenyl adenosine	1.07 ± 0.01 DE	1.32 ± 0.03 CD	1.02 ± 0.19 E	0.64 ± 0.02 F	0.75 ± 0.07 F	1.57 ± 0.08 C	1.01 ± 0.05 E	0.51 ± 0.04 F	2.66 ± 0.26 A	2.52 ± 0.14 A	0.7 ± 0.04 F	2.26 ± 0.1 B
Trans-zeatin-riboside	0.75 ± 0.03 BC	0.89 ± 0.07 B	0.7 ± 0.11 BCD	0.51 ± 0.03 DE	0.39 ± 0.05 E	0.45 ± 0.18 E	0.59 ± 0.09 CDE	0.54 ± 0.1 CDE	1.75 ± 0.11 A	1.92 ± 0.11 A	0.47 ± 0.05 E	0.90 ± 0.03 B
Kinetin	0.8 ± 0.22 A	0 ± 0 A	0.15 ± 0.13 A	0.11 ± 0.19 A	0 ± 0 A	0.14 ± 0.25 A	0.7 ± 0.24 A	0.09 ± 0.16 A	0.42 ± 0.73 A	0.26 ± 0.45 A	0.21 ± 0.36 A	0 ± 0 A
Methyljasmonate	25.46 ± 9.2 B	41.85 ± 15.91 A	7.18 ± 1.84 C	0 ± 0	0 ± 0	0 ± 0	30.96 ± 9.29 AB	0 ± 0	0 ± 0	0 ± 0	0 ± 0	0 ± 0
N-Jasmonic acid isoleucine	704.08 ± 218.74 A	422.87 ± 67.59 B	225.95 ± 37.27 C	33.18 ± 12.74 D	9.79 ± 7.01 D	0 ± 0 D	239.24 ± 43.57 C	36.9 ± 2.17 D	5.52 ± 1.06 D	16.28 ± 9.51 D	0 ± 0 D	6.8 ± 0.93 D
(±)-Jasmonic acid	365.15 ± 99.11 AB	213.45 ± 36.31 BCD	403.06 ± 96.53 A	163.91 ± 36.25 CDE	89.9 ± 43.19 DE	0 ± 0 E	269.16 ± 26.11 ABC	25.25 ± 4.49 E	0 ± 0 E	378.77 ± 185.82 AB	47.62 ± 11.82 DE	131.88 ± 27.96 CDE
Salicylic acid	5936.83 ± 1287.12 A	4600.56 ± 621.4 BC	3483.4 ± 382.85 CD	2661.81 ± 429.51 D	3373.65 ± 175.1 CD	2946.54 ± 554.53 D	5209.66 ± 456.41 AB	3505.46 ± 197.15 CD	4962.73 ± 319.12 AB	3085.64 ± 74.07 D	3140.04 ± 78.2 D	2969.25 ± 245.35 D
Abscisic acid	299.18 ± 39.15 ABCD	255.36 ± 92.27 BCD	204.43 ± 38.08 CD	156.61 ± 30.43 D	222.18 ± 15.68 CD	170.41 ± 18.89 D	303.49 ± 29.26 ABCD	213.36 ± 37.2 CD	346.35 ± 45.81 ABC	406.39 ± 167.28 AB	446.36 ± 61.02 A	355.33 ± 15.02 ABC

Notes: Mean ± SD; different capital letters indicate significant differences between treatments or stages according to one-way ANOVA at *p* < 0.01.

## Data Availability

All the generated or analyzed data are provided in this article and its Appendix A. The genes regulated by different light qualities are in Appendix A. Thirty-six raw Illumina-sequencing data have been deposited at the NCBI and can be accessed at the NCBI via accession number SAMN38816575-SAMN38816610.

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
