# Peer review of "Changes in Phytohormones and Transcriptomic Reprogramming in Strawberry Leaves under Different Light Qualities"

_ijms, 2024, doi:10.3390/ijms25052765_

Round 1

Reviewer 1 Report

Comments and Suggestions for Authors

The submitted manuscript “Interaction of phytohormone and transcriptomic reprogramming in strawberry leave under different light quality” by authors Li et al. is focused on investigation the quality and quantity of light on phytohormone and gene expression alongside with pigment content and photosynthetic performance of in leaves of strawberry plants, grown in greenhouse.

I have the following critical comments:

-        English language style and gramma needs intensive improvement!

Ex.: everywhere in the text “leave” is to be corrected. The singular of the word is “leaf” and the plural is “leaves”. In the way it is in the text “leave” it means the verb.

Line 18-19 – to be reformulated.

Line 24 – “with genes in antenna proteins” – to be reformulated.

Line 60-72 – sentences are too long, can be divided. L 60 – “Phytohormone participate” to be changed to either “Phytohormones participate” or “Phytohormone participates”. L 66 – “maintaine” to be changed to “maintains”. L 74 – “gene” and “pathway” – to be in plural. L 80 – “genes in photosynthesis-antenna proteins” to be changed to “genes of photosynthesis-antenna proteins”.

Results section

-        It is not described neither in this section nor in Materials and Methods what is indicated by K1 and K2, L1 and L2 etc. Probably the authors have in mind the two time periods of treatment – 3th and 7th day of treatment. This information should be included.

-        Line 91-96 – Abbreviation of “chlorophyll” – it is generally accepted to be abbreviated “Chl a” and “Chl b” but not “Ca and Cb”.

-        Line 99-100 – The statement that “reaction center of photosystem II is completely open, which is related to chlorophyll concentration” is not correct. Yes, Fo is registered at very low modulated light intensity (0.120 µmol photons m-2 s-1) when all PSII centers are open and all electron acceptors are in oxidized form, but this parameter is informative for the activity of reaction center of PSII (J. Cao, Govindjee, Biochim. Biophys. Acta 1015 (1990) 180–188, N.G. Bukhov, S.Ch. Sabath, P. Mohanty, Photosynth. Res., 23 (1990), pp. 81-87).

-        Tables 1 and 2 and figures except 1 are difficult to follow as the letters and numbers included are very small. The caption of Table 1 “Photosynthesis-related indices of strawberry leave” to be corrected for clear indication what is presented. The description of statistics to be included in the caption. In addition, it is not clear what represents the number after “±” – it is the SD or SE and both values, mean and after ± sigh is one bellow the other.

-        Line 109-123 – both paragraphs are very long. Can be divided into at least two.

-        Line 136 – The Table 2 legend – “Table 2. phytohormones levels of strawberry leave.” to be changed to “Table 2. Phytohormones levels in strawberry leaves” and the information about statistics to be included here.

-        Fig. 1 is OK, but in the legend, where are indicated the panels, it is not 1, 2 and 3 but a, b and c. To be indicated what is “padj”.

-        Fig. 2 – The figure legend needs more details and to mention how the colors correlate with strength.

-        Line 176 – “content of F0 exhibited” and line 177 “content of Pn” – it is not content but level.

-        Fig. 3 – the quality of the figure is very low. In the legend is not indicated panel a. The same hold true for Fig. 4.

Discussion section.

-        The whole discussion is generally repeating the results and not a real discussion is provided. A Conclusion section can be included to summarize the findings.

-         

Materials and Methods.

-        In the whole section there is no one reference cited. I hardly believe that the authors invented all the methodology.

-        Line 459 – The statement the N states for “no light” is not correct. The plants were treated with half of the day light intensity. Nowhere is given the light intensity – of the original day light and of blue and red light.

-        In subsection 4.2. is not given the way the water content was calculated. As I understand only the FW and DW were detected. For calculation of RWC you in general need the turgescent water content.

-        Line 473 – “values at 665nm and 649nm under the spectrophotometer” – the absorbance is detected by a spectrometer.

-        The formulas and coefficients for calculation of chlorophyll content – it is not clear from where are taken? There are different methods for calculations and the coefficients are dependent on the respective solvent used for extraction of pigments. For the concentration of chlorophyll is indicated “mg/g” but it is not clear per fresh or dry weight.

-        Line 479, 481, 484 – “inside to the outside” what the authors have in mind. It is not clear. Detection of the photosynthetic parameters by registration of Chlorophyll a fluorescence is performed usually on the adaxial (upper) surface of the leaf.

-        Line 487 – “phytohormone” should be in plural.

-        Line 488 – The leaf material is not “resuspended” in l. N2, it is not a liquid, but rather homogenized”.

-        Line 489 – “which containing mixed” to be changed to “which contains mixed”.

-        Line 488-493 – can be divided into 2 sentences.

-        A subsection describing the statistics is missing.

-        It is not indicated how many independent experiments were performed, how many samples were gathered at every timepoint and how many parallel samples were measured.

VThe submitted manuscript “Interaction of phytohormone and transcriptomic reprogramming in strawberry leave under different light quality” by authors Li et al. is focused on investigation the quality and quantity of light on phytohormone and gene expression alongside with pigment content and photosynthetic performance of in leaves of strawberry plants, grown in greenhouse.

I have the following critical comments:

-        English language style and gramma needs intensive improvement!

Ex.: everywhere in the text “leave” is to be corrected. The singular of the word is “leaf” and the plural is “leaves”. In the way it is in the text “leave” it means the verb.

Line 18-19 – to be reformulated.

Line 24 – “with genes in antenna proteins” – to be reformulated.

Line 60-72 – sentences are too long, can be divided. L 60 – “Phytohormone participate” to be changed to either “Phytohormones participate” or “Phytohormone participates”. L 66 – “maintaine” to be changed to “maintains”. L 74 – “gene” and “pathway” – to be in plural. L 80 – “genes in photosynthesis-antenna proteins” to be changed to “genes of photosynthesis-antenna proteins”.

Results section

-        It is not described neither in this section nor in Materials and Methods what is indicated by K1 and K2, L1 and L2 etc. Probably the authors have in mind the two time periods of treatment – 3th and 7th day of treatment. This information should be included.

-        Line 91-96 – Abbreviation of “chlorophyll” – it is generally accepted to be abbreviated “Chl a” and “Chl b” but not “Ca and Cb”.

-        Line 99-100 – The statement that “reaction center of photosystem II is completely open, which is related to chlorophyll concentration” is not correct. Yes, Fo is registered at very low modulated light intensity (0.120 µmol photons m-2 s-1) when all PSII centers are open and all electron acceptors are in oxidized form, but this parameter is informative for the activity of reaction center of PSII (J. Cao, Govindjee, Biochim. Biophys. Acta 1015 (1990) 180–188, N.G. Bukhov, S.Ch. Sabath, P. Mohanty, Photosynth. Res., 23 (1990), pp. 81-87).

-        Tables 1 and 2 and figures except 1 are difficult to follow as the letters and numbers included are very small. The caption of Table 1 “Photosynthesis-related indices of strawberry leave” to be corrected for clear indication what is presented. The description of statistics to be included in the caption. In addition, it is not clear what represents the number after “±” – it is the SD or SE and both values, mean and after ± sigh is one bellow the other.

-        Line 109-123 – both paragraphs are very long. Can be divided into at least two.

-        Line 136 – The Table 2 legend – “Table 2. phytohormones levels of strawberry leave.” to be changed to “Table 2. Phytohormones levels in strawberry leaves” and the information about statistics to be included here.

-        Fig. 1 is OK, but in the legend, where are indicated the panels, it is not 1, 2 and 3 but a, b and c. To be indicated what is “padj”.

-        Fig. 2 – The figure legend needs more details and to mention how the colors correlate with strength.

-        Line 176 – “content of F0 exhibited” and line 177 “content of Pn” – it is not content but level.

-        Fig. 3 – the quality of the figure is very low. In the legend is not indicated panel a. The same hold true for Fig. 4.

Discussion section.

-        The whole discussion is generally repeating the results and not a real discussion is provided. A Conclusion section can be included to summarize the findings.

-         

Materials and Methods.

-        In the whole section there is no one reference cited. I hardly believe that the authors invented all the methodology.

-        Line 459 – The statement the N states for “no light” is not correct. The plants were treated with half of the day light intensity. Nowhere is given the light intensity – of the original day light and of blue and red light.

-        In subsection 4.2. is not given the way the water content was calculated. As I understand only the FW and DW were detected. For calculation of RWC you in general need the turgescent water content.

-        Line 473 – “values at 665nm and 649nm under the spectrophotometer” – the absorbance is detected by a spectrometer.

-        The formulas and coefficients for calculation of chlorophyll content – it is not clear from where are taken? There are different methods for calculations and the coefficients are dependent on the respective solvent used for extraction of pigments. For the concentration of chlorophyll is indicated “mg/g” but it is not clear per fresh or dry weight.

-        Line 479, 481, 484 – “inside to the outside” what the authors have in mind. It is not clear. Detection of the photosynthetic parameters by registration of Chlorophyll a fluorescence is performed usually on the adaxial (upper) surface of the leaf.

-        Line 487 – “phytohormone” should be in plural.

-        Line 488 – The leaf material is not “resuspended” in l. N2, it is not a liquid, but rather homogenized”.

-        Line 489 – “which containing mixed” to be changed to “which contains mixed”.

-        Line 488-493 – can be divided into 2 sentences.

-        A subsection describing the statistics is missing.

-        It is not indicated how many independent experiments were performed, how many samples were gathered at every timepoint and how many parallel samples were measured.

In general, the manuscript needs intensive improvement in respect to clearly formulation the aim of investigation, presentation of data in Tables and Figures, presenting the results, discussing the data, improving the Materials and Methods section, improving the English language style and gramma.

Comments on the Quality of English Language

English language needs extensive style and gramma impruvement.

Reviewer 2 Report

Comments and Suggestions for Authors

Despite topic about link of different light quality with plant bgriowth is importnat, in the current form I can not evaluate this paper.

No any citation in the text. Many wrong statements. ASll senteces is very long and cinfused.

 Authors must completely re-write text with clear sentences, include citation and resubmit in readable form.

Lines 8- 10_ not appropriate sentence.

Mixing of several messages. Moreover, all plants require light for growth. It is well-know fast.

Lines 10- 12: Split two sejntence, please.

Line 16: “phytohormone results”?? If you mean “level” you need to mention.

Line 18: “differentail genes”??? what do you mean?

DEG is differentialy expressed genes.

 Line 22: unclear.

Line24:  “phytohormone interacts with genes in antenna proteins, carbon fixation” ¿??? Hormone can not interact with genes!

Lines 32- 44: weher are citation? How can you describe all bin one sentence? It is not reaabale.

Whole introduction: no any citations, very long sentence , wrong statetements:

Line 79 – 83: in the current form sentence does not have any sense.

Line 177: “The content of Pn exhibited” – what is the contents of Pn?

Comments on the Quality of English Language

In the current form text is not readable.

Round 2

Reviewer 1 Report

Comments and Suggestions for Authors

The revised version of manuscript ”Changes in phytohormones and transcriptomic reprogramming in strawberry leaves under different light qualities” by authors Li et al was significantly improved but there are still things that require attention.

-          Line 15 and everywhere in the text – abbreviation of chlorophyll “chl b” should be consistently presented – everywhere to be with capital letter “Chl a, Chl b”.

-          F0 – everywhere in the text to be in subscript – “F0

-          Line 35 – “Blue light (660 nm) and red light (460 nm)” – it is the other way round.

-          Line 100 – “N2L” to be changed to “NL2”.

-          In the whole subsection 2.2. are used multiple times “decrease and decrease”. Should be substituted by synonyms.

-          The last sentence of every subsection in Discussion is one and the same. Should be reformulated.

-          Line 520 – in the formula for water content something is missing. It is not clear whether it was RWC (for determination of which is needed the turgescent weight as well). In the presented formula only DW is given.

-          The reference given for determination of Chl a and Chl b (72, Wu Q. 2018) is not correct. I do not believe that this author determined the coefficients of absorbance at different wavelengths of Chl a and Ch b in 95% ethanol and constructed the formulas. The formulas are very similar to that reported by Linchenthaler H.K., 1987 in Methods of Enzymology. This paper is considered basic and classic for the determination of pigment content and is cited in majority of papers reporting on extraction and determination of pigment content. The authors should check in reference ( 72) what is the original formula and cite it instead of a paper that just used it for calculations of one’s results.

Comments on the Quality of English Language

The manuscrip still needs corrections in respect to methodology, citing and styling.

Reviewer 2 Report

Comments and Suggestions for Authors

Thank you, it is better , but so many corrections are required.

Some details:

Line 56: contents can not improve, but can only increase. Moreover, authorsneed to explain what they mean ascontents in fruit. Fruit have so many cell type and possibility of auxin distribution. Moreover, contents is consensus of so many factors: high contents can related with high synthesis, low efflux, and combination of both. In relaity contents does nit have too much biological sense.

Lines 61- 62: SA mentioned on 61, but explanation on 62.

Lines 62- 66: long sentences, very confused.

Lines 71- 79: M&M.

Lines 80-86: please. Re-write more clear.

Line 90: “and resistance of plants.“ ?? What is resistance of plants?

Lines 91- 93: what is LL1 and LL2? And other abbreviations??

Line 177: „significant positive correlation with the genes“ ? Maybe gene expressions??

Line 188: „significant negative correlation with the genes“ ? Gene expressions?

M&M require significant polishing. Some non-complete sentences etc.

Lines 305-332: There are many YUCCA genes , which expressed in different location and have different roles. It must be mentioned.

Comments on the Quality of English Language

Many complicated sentences, some non complete.
